# Evaluation of Effects of Spray Drying Conditions on Physicochemical Properties of Pomegranate Juice Powder Enriched with Pomegranate Peel Phenolic Compounds: Modeling and Optimization by RSM

**DOI:** 10.3390/foods12102066

**Published:** 2023-05-20

**Authors:** Jouhaina Hadree, Fakhri Shahidi, Mohebbat Mohebbi, Mohammadreza Abbaspour

**Affiliations:** 1Department of Food Science and Technology, Faculty of Agriculture, Ferdowsi University of Mashhad, Mashhad 91779-48974, Iran; 2Department of Food Science, Faculty of Agriculture, University of Aleppo, Aleppo 12212, Syria; 3Targeted Drug Delivery Research Center, School of Pharmacy, Mashhad University of Medical Sciences, Mashhad 91388-13944, Iran

**Keywords:** pomegranate, phenolic extract, spray drying, RSM, optimization, powder

## Abstract

In this study, the effects of pomegranate peel extract concentration (2.5–10%), drying temperature (160–190 °C), and feed flow rate (0.6–1 mL/s) on the properties of pomegranate juice powder enriched with pomegranate peel phenolic compounds and produced by spray drying were investigated. The moisture content, water activity (a_w_), solubility, water absorption capacity (WAC), hygroscopicity, dissolution time, total phenolic content (TPC), Carr index (CI), Hausner ratio (HR), and brightness (L*) of the samples were evaluated, and the optimal powder production conditions were obtained using response surface methodology (RSM). The results showed that the optimal conditions were found to be the phenolic extract concentration of 10%, the drying temperature of 189.9 °C, and the feed flow rate of 0.63 mL/s, considering the minimization of the moisture content, a_w_, hygroscopicity, dissolution time, CI, HR, and L*, as well as the maximization of solubility, WAC, and TPC. The effect of the phenolic extract concentration was very significant (*p* < 0.01) on the WAC, hygroscopicity, dissolution time, TPC, CI, HR, and L* of the powder. Moreover, the effect of the drying temperature was very significant (*p* < 0.01) on the a_w_, hygroscopicity, dissolution time, CI, and HR of the powder and significant (*p* < 0.05) on its moisture content. The effect of the feed flow rate was very significant (*p* < 0.01) on the solubility, hygroscopicity, and dissolution time of the powder and significant (*p* < 0.05) on its moisture content. Therefore, we found that the spray drying conditions, such as high temperature, did not negatively affect the content of phenolic compounds in pomegranate powder, and the physical properties of the resulting powder were acceptable. Thus, pomegranate powder enriched with phenolic compounds can be used as a food additive or as a dietary supplement for medicinal use.

## 1. Introduction

Fruits are seasonal like other agricultural products. Most of them have a short shelf-life due to high perishability. A significant portion (about 30%) of fruits is lost from harvest to consumption due to various reasons. Therefore, their shelf-life should be extended by using different methods [1].

Pomegranates (*Punica granatum*) belong to the *Punicaceae* family. Due to the fact that pomegranates are native to the Middle East, the largest cultivated area in the world belongs to this region [2]. Pomegranate fruits contain acids, sugars, vitamins, polysaccharides, polyphenols, and important minerals [3]. The presence of phenolic compounds such as anthocyanins, ellagic acid, phytoestrogen flavonoids, and tannins in different parts of pomegranates has been confirmed [4]. Pomegranates have received considerable attention due to their phenolic compounds, free radical-scavenging activity, and antioxidant, immunity, antiviral, antimicrobial, antiparasitic, and anticancer properties [5].

In recent years, the phenolic compounds of pomegranate peels have drawn attention due to their health benefits, such as antioxidant activity [6]. Considering that the pomegranate peel, as a byproduct, constitutes about 40–50% of the total weight of the fruit [7] and has some biological activities, it can be considered a potential source of bioactive compounds used in the food industry [8]. There are many reports about the strong antioxidant and antibacterial activities of pomegranate peel extract, which are related to the high value of the phenolic compounds present in this part [9].

Spray dryers are used to obtain dry powder from liquid materials [10]. In these dryers, the liquid is transferred from the feed tank to the nozzle by the pump. Then, it is sprayed in the form of droplets into the dryer chamber. After that, it makes contact with hot air, causing the rapid evaporation of water [11]. The physical properties of the produced powders are very important and depend on technological parameters such as the feed flow rate, temperature, nozzle type, dry air flow rate, atomizer speed, and type and concentration of the carrier agent. Owing to the usability of fruit juice powders in different time conditions, their easy transfer, and their numerous uses in the food, pharmaceutical, and chemical industries as raw materials, researchers tend to investigate them [12]. Maltodextrin is a starch hydrolysis product with a dextrose equivalent (DE) of less than 20 and high solubility in water. It is commonly used as a bulking agent, stabilizer, and thickener in the food and beverage industries. In powder production, maltodextrin is used as a carrier or filler for dry ingredients to improve flowability and prevent clumping [13].

According to changing lifestyles, especially in developed and developing countries, there is a great demand for a wide range of powdered food products with an emphasis on quality [14], and because the phenolic compounds can be a potential ingredient for functional food and nutraceutical formulations, further studies on the bioavailability and efficacy of agro-industrial byproducts, such as pomegranate peels in improving health outcomes, are warranted [15].

In previous studies, the utilization of pomegranate peel extract in films, coating formulations, ice cream, dairy products, apple juice, carrot juice, jellies, cookies, and meat products has been investigated [16,17,18,19,20,21,22,23,24]. However, limited studies on the application of pomegranate peel powder have been reported, and it is still underutilized in food systems; so the purpose of this research, because of the presence the huge amount of juice industry wastage and based on antioxidant and antimicrobial properties of the pomegranate peel extract, is enriching pomegranate juice with phenolic compounds obtained from pomegranate peel extract, investigating the effect of spray drying conditions on the physicochemical properties of the powder, and determining the optimal parameters using RSM.

## 2. Materials and Methods

### 2.1. Materials

The materials used in this study are as follows: pomegranate concentrate (Alis, Iran), maltodextrin (DE = 16) (FIC, China), NaCl, Folin–Ciocalteu reagent (Merck, Germany), sodium carbonate (Sigma-Aldrich, Germany), gallic acid (Genay, France), 2,2-diphenyl-1-picrylhydrazyl (DPPH) (Sigma, Germany), and distilled water.

### 2.2. Methods

#### 2.2.1. Preparation of Pomegranate Juice

Pomegranate concentrate (62° Brix and without additional matter) was purchased from Alis Co., (Mashhad, Iran) and stored at 4 °C. The Brix of the concentrate was adjusted to 10 for preparing the final solution.

#### 2.2.2. Preparation of Pomegranate Peel Extract

Fresh pomegranate fruits (*Punica granatum* L.) were supplied from local gardens (Mashhad, Iran) and stored at 6 °C prior to experiments. They were rinsed with water and manually peeled. The peels were separated and dried in a vacuum oven at 40 °C for 2 days. The dried peels were then pulverized by a domestic grinder. After passing through a screen with a mesh size of 40, the powder was transferred to a glass bottle covered with aluminum foil and kept at room temperature until use. Next, 5 g portions of the finely-powdered peels were mixed with 150 mL water. The mixtures were subsequently stirred for 4 h at 55 °C prior to centrifugation at 7000× *g* for 20 min at 4 °C [15]. After that, for the concentration of 2.5% (dry basis), the extract was directly used, whereas, for the concentrations of 6.25 and 10% (dry basis), the extract was concentrated using a rotary evaporator at 50 °C until its volume would reach that of 2.5%.

#### 2.2.3. Spray Drying

The powder was produced using a mini spray dryer (BUCHI, B-290, Laboratory-Techniques LTD, Flawil, Switzerland). The spray dryer operates concurrently and has a spray nozzle and a two-fluid atomizer with an orifice that is 0.7 mm in diameter. Spray drying, on the basis of pre-treatment, was carried out at various inlet air temperatures (160–190 °C) and feed flow rates (0.6–1 mL/s) with constant aspirator rate (0.6 mL/s) and pressure (4.5 bar). Once the juice’s total solids were adjusted at 10% (*w*/*v*), the following substances were added, including maltodextrin (DE = 16) at a ratio of 1:1 with the dry matter of the concentrate and the pomegranate peel extract at 2.5–10% (dry basis). Finally, the solution was spray-dried.

### 2.3. Experiments

#### 2.3.1. Moisture Content

The moisture content of the pomegranate powder was determined using the oven method [25].

#### 2.3.2. Water Activity

a_w_ was determined using an a_w_-meter (Rotronic Hygrolab2, Bassersdorf, Switzerland) at 25 °C.

#### 2.3.3. Solubility

A total of 100 mL of distilled water was poured into a glass container. Then, 1 g of the powder was carefully weighed and added to the container at room temperature. The resulting solution was immediately agitated by an Ultra-Turrax homogenizer at 15,000× *g* rpm for 3 min. After that, it was centrifuged (Sigma, 3–30 k model, Germany) at 3000× *g* rpm for 5 min. The supernatant was carefully poured into clean, dry plates that had already been weighed and dried in an oven at 105 °C for 5 h to reach a constant weight. The weight of the dry matter was the basis for calculating the powder solubility [26].

#### 2.3.4. Water Absorption Capacity

A total of 1 g of the powder was dissolved in 30 mL of distilled water and kept at room temperature for 18 h, followed by being centrifuged (Sigma, 2-16KC, Osterode am Harz, Germany) at 3400× *g* for 34 min. Finally, WAC was calculated from the volume difference [27].
WAC(mL)=Vw−Vs
where Vw is the volume of added water (mL), and Vs is the volume of supernatant (mL).

#### 2.3.5. Hygroscopicity

A total of 1 g of the powder was weighed in a clean, dry plate with a certain weight, which was immediately placed in a desiccator containing a saturated solution of water and salt. After capping, the desiccator was placed in an incubator at 25 °C for one week. Afterwards, the plate was weighed. Finally, the amount of water absorption was calculated as g of water absorbed in 100 g of dry matter (g/100 g) [28].
$$Hygroscopicity=\frac{\Delta m/\left(m+m1\right)}{1+\left(\Delta m/m\right)}*100$$
where ∆*m* is the increasing of powder’s weight after reaching equilibrium (g), *m* is the powder’s initial mass (g), and *m*1 is the initial free water content of the powder before being exposed to air’s humidity (g/100 g).

#### 2.3.6. Dissolution Time

A total of 50 mg of the powder was carefully weighed by a digital scale in a small test tube (mini-tube). Then, 1 mL of distilled water was added, and the tube was placed on a vortex mixer (VELP Scientifica, Usmate Velate (MB), Italy) with 50% of the vortex power. Finally, the complete dissolution time of the powder was recorded in seconds for each sample [29].

#### 2.3.7. Total Phenolic Content

The TPC of the samples was measured using the Folin–Ciocalteu method [30]. Results were expressed as mg gallic acid equivalents (GAE)/g dry matter.

#### 2.3.8. Carr Index and Hausner Ratio

For calculating the CI and HR, the bulk (*Bd*) [31] and tapped (*Td*) [32] densities were determined.

The flowability and cohesiveness of the powder were computed using the CI [33] and HR [34] by the following equations:$$\mathrm{CI}\left(\%\right)=100*\left[1-\frac{Bd}{Td}\right]$$
$$\mathrm{HR}=\frac{Td\;}{Bd}$$
where *Bd* is the bulk density (g/mL), and *Td* is the tapped density (g/mL).

#### 2.3.9. Brightness

The L* of the pomegranate powder samples was determined using image-capturing equipment, including a dark chamber and two fluorescent lamps. The images were captured using a Canon power shot 1000 D camera connected to a computer through a USB port. The camera was fixed at a distance of 20 cm, and the images were analyzed by Imagej version 1.47 [35].

#### 2.3.10. DPPH Free Radical-Scavenging Activity

The ability of optimal powder to act as a free radical scavenger was evaluated by the method of [36].
Inhibition=(Ac−As)/Ac)∗100
where

*Ac*: absorbance of the blank (with the same chemicals, except the sample)

*As*: absorbance of the sample.

#### 2.3.11. Experimental Design and Statistically Analysis

Designing the experiments, analyzing the results, and determining the optimal conditions of the powder production process were done using RSM and Design Expert version 13.0.0 (Statease Inc., Minneapolis, MN, USA). In this research, the face-centered central composite design (FCCD) was used with three independent variables in three levels and six replications for the central point of the design, so that the total number of treatments was equal to 20. Due to the fact that some systematic and undetectable errors may occur in the observed responses, the test is repeated at the central point of the design to reduce the errors that may occur [37]. The independent variables included the phenolic extract concentration (2.5–10%), drying temperature (160–190 °C), and feed flow rate (0.6–1 mL/s). Furthermore, the dependent variables (response) included the moisture content (%), a_w_, solubility (%), WAC (mL), hygroscopicity (%), dissolution time (s), TPC (mg/100 g), CI (%), HR, and L*.

To evaluate the impacts of the factors on the responses, the quadratic polynomial equation was fitted to the empirical data:$$Y=\beta_{0}+\sum_{i=1}^{3\;}\;\beta_{i\;}X_{i\;}+\sum_{i=1}^{3}\beta_{ii}\;X_{i}^{2}+\sum_{i=1}^{2}\sum_{j=i+1}^{3}\beta_{ij\;}\;X_{i\;}X_{j}$$
where Y is the estimated response; *β*_0_, *β_i_*, *β_ii_*, and *β_ij_* are constant coefficients; and *X_i_* and *X_j_* express the independent variables. The quality and accuracy of the regression model and the appropriateness of the fit were determined by parameters such as lack of fit and coefficient of determination (R^2^) [38].

## 3. Results and Discussion

### 3.1. Model Fitting

The experimental results were used to optimize the production conditions of the pomegranate juice powder enriched with the phenolic extract of pomegranate peels (Table 1). In order to study the correlation between the response and independent variables and predict each of the responses, including moisture content (%), a_w_, solubility (%), WAC (mL), hygroscopicity (%), dissolution time (s), TPC (mg/100 g), CI (%), HR, and L*, different models were fitted to the empirical data.

According to the results of the analysis of variance (ANOVA), the quadratic model was the best one to explain the effects of the independent variables on moisture content, solubility, WAC, and L*; the linear model to describe the effects of the independent variables on hygroscopicity, dissolution time, and TPC; and the 2FI model to investigate the effects of the independent variables on a_w_, CI, and HR. These models were the most significant and had the highest R^2^ values and the lowest standard deviations and predicted residual sum of squares (PRESS).

### 3.2. Effects of Independent Variables on Properties of Powder

#### 3.2.1. Moisture Content

Understanding the moisture content is crucial for powders as it affects various aspects such as stability, flowability, drying efficiency, stickiness, bioactive agent oxidation, and microbial growth [39].

The ANOVA results (Table 2) showed that the effects of drying temperature and feed flow rate were significant (*p* < 0.05) on the moisture content of the powder. Furthermore, the interactive effects of phenolic extract concentration-drying temperature and phenolic extract concentration-feed flow rate were significant (*p* < 0.05); and the interaction between drying temperature and feed flow rate was highly significant (*p* < 0.01). In addition, the quadratic terms of the phenolic extract concentration, drying temperature, and feed flow rate were very significant (*p* < 0.01).

By introducing the quadratic model into the backward elimination algorithm, the following equation was obtained, composed of the coded factors that had significant effects on the moisture content of the powder:Moisture content = +10.60 + 0.2612A − 0.3440B + 0.2936C + 0.3573AB + 0.3701AC − 0.5259BC − 0.8894A² + 0.7612B² − 1.17C²(1)
where A is the phenolic extract concentration (%), B denotes the drying temperature (°C), and C represents the feed flow rate (mL/s). From Equation (1), it can be concluded that the reducing effect of the drying temperature was the most profound one on the moisture content of the powder.

The moisture content of the pomegranate powder enriched with the pomegranate peel phenolic extract produced by spray drying was between 7.84 and 11.39%, as shown as in Table 1. The 3D contour plots of the effects of the phenolic extract concentration, drying temperature, and feed flow rate on the moisture content are depicted in Figure 1. It was realized that the drying temperature produced a decreasing effect up to 178 °C, which means that increasing the drying temperature led to a decrease in the moisture content, which reached a plateau at higher temperatures. This is due to the fact that by increasing the temperature, the heat transfer rate from air to the sprayed particles would be higher, and as a result, the evaporated water quantity increased. This result is in agreement with the studies of Horuz et al. [31], Fazaeli et al. [40], Goula and Adamopoulos [41], and Kha et al. [1], who respectively found that increasing the inlet air temperature in the drying of unclarified pomegranate juice, black mulberry juice, orange juice concentrate, and Gac fruit aril powder resulted in a drop in moisture content values. The reason for the slight increase in the moisture content at higher drying temperatures could be the formation of a dry layer on the surface of the droplets, which prevented water evaporation. Chegini and Ghobadian [42] and Jumah et al. [43] concluded that the moisture contents of orange juice powder and spray-dried jameed increased at high inlet air temperatures. Goula et al. [44] studied the production of tomato powder and declared that an increase in the inlet air temperature led to a lower moisture content, except for the increase from 130 to 140 °C at drying air flow rates of 21.00 and 22.75 m^3^/h.

Furthermore, as depicted in Figure 1, there was an increase in the phenolic extract concentration up to approximately 7% increased moisture content, which subsequently decreased. The initial increase in the moisture content may be due to the morphological properties and interactions of the particles, which probably caused a rise in the attractive forces between them. The reduced moisture content values at the high concentration of the phenolic extract could be explained by the increased total solids content, which reduced the available water to be evaporated. Moreover, the high molecular weight of the phenolic compounds in the extract could be one of the reasons behind the declined moisture content. In separate studies on Gac fruit aril powder and watermelon powder, Kha et al. [1] and Quek et al. [45] claimed that an increase in total solids content by adding maltodextrin resulted in a decrease in the moisture content of the powder. Ong et al. [46] also concluded that the low molecular weight of sugar was one of the reasons for absorbing water from the air and, consequently, increasing the moisture content. Akhtar et al. [47] indicated that the phenolic compounds present in pomegranate peel included anthocyanins, gallotannins, hydroxycinnamic acids, hydroxybenzoic acids, and hydrolyzable tannins, i.e., ellagitannins, and gallagyl esters. Kharchoufi et al. [14] stated that the methanolic and aqueous pomegranate peel extracts had similar phenolic profiles, but the presence of these high-molecular-weight phenolics, namely ellagitannins, proanthocyanidins, complex polysaccharides, and flavonoids, caused a difference in the properties of the extracts.

Considering Figure 1, it can be said that the feed flow rate up to 0.8 mL/s had an increasing influence on the moisture content of the powder, which followed a decreasing trend thereafter. Large and coarse droplets can result from the formation of a hard surface layer, leading to an initial increase in moisture content. Similar results have been observed by Chegini and Ghobadian [42] and Jumah et al. [43] in the studies on spray-dried orange juice powder and spray-dried jameed. Moreover, Mohanta et al. [48] explained that the insufficient heat transfer was caused by the disproportion between the feed flow rate and the volume of the solution present in the chamber. Therefore, the decrease in the moisture content at high feed flow rates could be due to the influence on the shape and size of the particles. Mestry et al. [10] indicated that increasing the feed flow rate at high atomizer pressures produced fine droplets in the fermented mixed juice of carrot and watermelon, which led to a decrease in the particle diameters and reduced the moisture content.

#### 3.2.2. Water Activity

Water activity is a measure of free water in food that is responsible for biochemical reactions and is an important indicator in determining the microbial stability of food [49]. Water activity has a great impact on the shelf life, safety, texture, aroma, and taste of the product [50]. All microbial activities stop in water activity below 0.6 [51].

The ANOVA results (Table 3) indicated that the drying temperature had a very significant effect on a_w_ (*p* < 0.01), as well as the interaction between the phenolic extract concentration and feed flow rate, which had a highly significant effect (*p* < 0.01).

By introducing the 2FI model into the backward elimination algorithm, the following equation was obtained, composed of the coded factors that had significant effects on the a_w_ of the powder:a_w_ = +0.1791 + 0.0017A − 0.0078B + 0.0001C − 0.0005AB − 0.0145AC + 0.0005BC(2)

Equation (2) shows that the decreasing influence of drying temperature was the largest effect on the a_w_ of the powder.

The a_w_ values of the powder were between 0.154 and 0.201, revealing that the resulting powder was resistant to chemical changes and microbial spoilage. Quek et al. [45] reported that each food product with an a_w_ lower than 0.6 was considered microbiologically stable, and Marques et al. [52] claimed that each food with an a_w_ lower than 0.2 was considered chemically stable. However, the results showed that increasing the inlet air temperature led to a reduction in the a_w_ values, agreeing with the study on spray-dried kefir powder by Atalar and Dervisoglu [53].

Figure 2 denotes the 3D contour plots of the interaction between the phenolic extract concentration and feed flow rate, which indicates that a_w_ was raised at low extract concentrations by elevating the feed flow rate. This is in contrast to that at high extract concentrations where a_w_ was reduced by increasing the feed flow rate because of the rise in the soluble solids content. Quek et al. [45] and Kha et al. [1] mentioned that by increasing maltodextrin concentration, the a_w_ of the powder decreased. Galaz et al. [54] stated, in their study on drum-dried pomegranate peels, that the a_w_ values ranged from 0.36 to 0.391.

#### 3.2.3. Solubility

Solubility is one of the most important functional properties of food powders, which shows the behavior of the powders during regeneration in water. The solubility parameter is used as an indicator of the amount of dissolved or undissolved particles [55].

The ANOVA results (Table 2) indicated that the linear term of the feed flow rate had a very significant effect on the solubility of the powder (*p* < 0.01), Moreover, the interactive and quadratic terms of the three factors, including phenolic extract concentration, drying temperature, and feed flow rate, very significantly affected the response (*p* < 0.01).

By introducing the quadratic model into the backward elimination algorithm, the following equation was obtained, composed of the coded factors that had significant effects on the solubility of the powder:Solubility = +91.03 − 0.0085A + 0.0352B + 0.4574C + 0.3372AB − 0.3378AC + 0.3312BC + 0.746 A² − 0.4006B² − 0.4966C²(3)

Equation (3) demonstrates that the increasing influence of the feed flow rate was the most remarkable effect on the solubility of the powder.

As shown in Table 1, the solubility of the powder in water was higher than 89.5%. Figure 3 shows that the phenolic extract concentration initially had a reducing effect on the powder solubility, while it was reversed at concentrations higher than 6%. The initial decrease might be due to the fact explained by Nur Hanani et al. [15], who investigated the addition of pomegranate peel powder to fish gelatin films and mentioned that the powder had a significant decreasing effect on the solubility of the film. They suggested that the formation of covalent bonds, as a result of the cross-linking of phenolic compound and simple sugar, was one of the reasons for the decrease in the solubility of the film. Horuz et al. [31] evaluated the effect of spray drying on the pomegranate juice properties and cited that the high molecular weight of maltodextrin was a reason for the drop in the powder solubility. In our research the reason behind the rise in the powder solubility might have been due to more penetration of water into the powder particles; depending on the size, shape, and distribution of the particles; or due to the high-weight molecules of the phenolic compounds, which inhibited the stickiness of the particles and increased the solubility. Naji-Tabasi et al. [56] examined barberry juice powder and stated that the morphological shape of maltodextrin molecules was the reason for increasing the porosity and thus the powder solubility.

In addition, Figure 3 shows that the solubility of the powder increased by increasing the inlet air temperature. However, at temperatures higher than 184 °C, the powder solubility was reduced, which might have occurred as a result of the low density of the powder by increasing the temperature, which means that large particles prevented the flow of the particles on the surface of the water. On the other hand, at very high temperatures, the fast formation of a stiff surface layer on the particles restrained the penetration of water through them, thus decreasing the solubility of the powder in addition to the probability of destroying the surface part for water absorption. However, the solubility remained above 89.5%. Jafari et al. [57], Fazaeli et al. [40], and Karaaslan and Dalgıç [58] mentioned that the solubility of pomegranate juice powder, black mulberry juice powder, and licorice extract increased by increasing the drying temperature. Nevertheless, Horuz et al. [31], Chegini and Ghobadian [42], and Quek et al. [45] concluded that very high inlet air temperatures caused the solubility of pomegranate, orange, and watermelon juice powders to lower. They declared that very high inlet air temperatures resulted in the fast drying of the drop surface, constituting a stiff surface layer on the particles. This inhibited the diffusion of the water molecules into the powder particles, thus reducing the solubility.

Figure 3 also indicates that the feed flow rate had a positive effect on the powder solubility, signifying that which was reported by Chegini and Ghobadian [42], who worked on orange juice powder and maintained that by increasing the feed flow rate, the insoluble solids content decreased due to the higher moisture content of the powder particles or forming a thin surface layer on the drops. Additionally, increasing the feed flow rate could give rise to the droplet size, hence enlarging the particles with coarse surfaces, which aid the powder to dissolve in water; Mestry et al. [10] indicated a similar result in the investigation of the influence of spray dryer parameters on the properties of the fermented mixed juice of carrot and watermelon powder.

#### 3.2.4. Water Absorption Capacity

Measuring the water absorption capacity (WAC) involves adding water using centrifugation to the powder or food material and calculating the amount of the retained water in the pelleted material in the centrifuge tube [59]. WAC is related to the capability of dry food to absorb water, and it is exactly associated with the hydration capacity [60].

According to Table 2, it can be seen that the phenolic extract concentration had a very significant effect (*p* < 0.01) on the WAC of the powder. In addition, the interactive effect of the phenolic extract concentration and the drying temperature was significant (*p* < 0.0), and the interaction between the phenolic extract concentration and the feed flow rate had a highly significant effect (*p* < 0.01). Furthermore, the quadratic terms of the phenolic extract concentration and drying temperature were very significant (*p* < 0.01).

By introducing the quadratic model into the backward elimination algorithm, the following equation was obtained, composed of the coded factors that had significant effects on the WAC of the powder:WAC = +1.57 − 0.1150A − 0.0350B + 0.0040C + 0.0725AB + 0.1850AC − 0.0500BC + 0.2464A² + 0.1864B² − 0.0886C²(4)

It can be concluded from Equation (4) that the negative effect of the phenolic extract concentration was the most important one on WAC.

As shown in Table 1, the WAC values ranged between 1.46 and 2.25 mL. Figure 4 depicts the 3D contour plots of the phenolic extract concentration, drying temperature, and feed flow rate. It can be observed that increasing the phenolic extract concentration up to 6.5% led to a reduction in WAC. However, higher concentrations resulted in higher WAC values. However, WAC was very slightly affected by the inlet air temperature, and the results were approximately similar. The influence of the feed flow rate was also negligible. It should also be noted that the feed flow rate had a reducing effect on WAC at low phenolic extract concentrations, while it had a raising effect at high concentrations. The high molecular weight of the phenolic compound, which prevents the stickiness of particles, could be the reason for the increase in WAC. Shaari et al. [61] expressed that the WAC of pineapple powder produced by spray drying was 1.68 g due to maltodextrin properties, which prevented the stickiness of the powder particles.

#### 3.2.5. Hygroscopicity

Hygroscopicity is the tendency of a powder to absorb moisture from an environment with higher relative humidity and to achieve equilibrium with the humidity in the atmosphere [62]. Fruit powders, glucose, and fructose, due to their polar ends, lead to strong interactions with water molecules, which increases the hygroscopicity and can affect other quality properties of powders [63].

The results presented in Table 4 indicate that the three independent variables, including the phenolic extract concentration, drying temperature, and feed flow, had very significant effects (*p* < 0.01) on the hygroscopicity of the powder.

By introducing the linear model into the backward elimination algorithm, the following equation was obtained, composed of the coded factors that had significant effects on the hygroscopicity of the powder:Hygroscopicity = +26.83 − 0.4531A + 0.4112B − 0.5941C(5)

Equation (5) shows that feed flow rate was the most important factor in the hygroscopicity of the powder, which had a reducing effect on the response.

The 3D contour plots of the phenolic extract concentration, drying temperature, and feed flow rate are shown in Figure 5, indicating that by increasing both the phenolic extract concentration and feed flow rate, the powder hygroscopicity declined, but increasing the inlet air temperature elevated it. The high molecular weight of the phenolic substances could be the reason for the decrease in the hygroscopicity of the powder, because of their high glass transition temperature (T_g_). Moreover, the moisture content of the powder could have an impact on the powder’s ability to absorb the water molecules from the surrounding environment, resulting in the powder having less moisture content and more hygroscopicity. Caparino et al. [64] declared that adding maltodextrin was the reason behind the decrease in the hygroscopicity of spray-dried mango powder because of its high molecular weight. Tonon et al. [28] indicated that the low moisture content of acai powder led to an increase in its hygroscopicity. Goula et al. [44] showed that increasing the drying temperature reduced the hygroscopicity of tomato powder. Karaaslan and Dalgıç [58] assessed the spray drying of licorice extract and concluded that increasing the drying temperature gave rise to the T_g_ value and, as a result, lowered the hygroscopicity of the powder.

#### 3.2.6. Dissolution Time

The ease of regenerating food powder directly reflects the quality of the powder, especially instant powders. Therefore, powder properties such as solubility and dissolution time are used to describe food powder [65].

ANOVA results (Table 4) showed that the effects of the three independent variables were very significant (*p* < 0.01) on this response.

By introducing the linear model into the backward elimination algorithm, the following equation was obtained, composed of the coded factors that had significant effects on the dissolution time of the powder:dissolution time = +19.64 + 0.8800A + 0.3380B + 0.9300C(6)

From Equation (6), it can be concluded that the positive effect of the feed flow rate was the most dramatic one on the dissolution time of the powder.

The 3D contour plots of the effects of the phenolic extract concentration, drying temperature, and feed flow rate on the dissolution time are demonstrated in Figure 6. The response ranged from 17.15 s to 21.5 s. The results revealed that the three factors had increasing impacts, i.e., increasing each of the factors led to a rise in the dissolution time of the powder. This result was inconsistent with those of Goula and Adamopoulos [66], who stated that increasing the drying temperature raised the rehydration ability and decreased the dissolution time of tomato pulp powder. However, regarding the influences of the drying temperature and feel flow rate, the present study is, respectively, in agreement and contrast with that which was previously carried out by Chegini and Ghobadian [42], who concluded that the orange juice powder produced at higher inlet air temperatures, required more time for dehydration because of its lower moisture content. On the other hand, they cited that an increase in feed flow rate led to a decrease in the dissolution time of the powder. Goula et al. [44] conducted research on the effects of spray drying conditions on tomato powder properties and mentioned that increasing the drying temperature resulted in the elevation of the dissolution time of the powder. Caliskan and Dirim [67] spray-dried sumac extract and expressed that increasing the drying temperature resulted in an increase in the wettability time of the powder.

#### 3.2.7. Total Phenolic Content

The health benefits of phenolic compounds have been investigated in recent studies, and their potent antioxidant effects were proven [68].

According to the ANOVA results in Table 4, it can be said that only the phenolic extract concentration had a highly significant effect (*p* < 0.01) on TPC, whereas the other variables had no significant effect (*p* > 0.05).

By introducing the linear model into the backward elimination algorithm, the following equation was obtained, composed of the coded factors that had significant effects on the TPC of the powder:TPC = +3522.40 + 1087.70A + 42.60B − 88.60C(7)

Figure 7 illustrates the effect of the phenolic extract concentration on the TPC of the powder, which ranged between 2170 and 4976 mg gallic acid/100 g. Equation (7) explains that the phenolic extract concentration had a positive effect on TPC, meaning that increasing the concentration of the phenolic extract in the feed solution gave rise to the TPC values of the powder, which was expected due to the high percentage of the phenolic compounds in the extract. Dewanto et al. [69] showed that heat processing had no significant effect on the TPC of sweet corn, whereas Naji-Tabasi et al. [70] cited that heat treatment was an effective positive factor in the TPC of the barberry fruit pulp powder. Finally, it could be concluded that the spray drying process had no significant effect on the TPC of the powder, which conforms to that reported by Zea et al. [71] who claimed that vitamin C and betalain in pitaya powder were not affected by freeze drying. Desobry et al. [72] suggested that the large number of small particles helped protect β-carotene from oxidation, which is possible in the case of spray drying at high temperatures.

#### 3.2.8. Flowability and Cohesiveness

Flowability is the relative movement of the particles between themselves or along the surface of the vessel wall, measured as a Carr index, while cohesiveness is an internal property of the powder and an index of the retentive forces of the particle together, measured as a Hausner ratio. The lower the Carr index and Hausner ratio, the better the flow and the less cohesive the powder [10]. Different ranges for the Carr Index and the Hausner Ratio have been defined by Lebrun et al. [73] as presented in Table 5.

The ANOVA results in Table 2 revealed that the phenolic extract concentration and drying temperature had very significant effects on the CI and HR values of the powder (*p* < 0.01), Additionally, the interaction between the phenolic extract concentration and drying temperature was very significant (*p* < 0.01), and that of the drying temperature and feed flow rate was significant (*p* < 0.05) on the CI and HR of the powder.

By introducing the 2FI model into the backward elimination algorithm, the following equation was obtained, composed of the coded factors that had significant effects on the CI and HR of the powder:Carr index = +24.44 + 3.45A − 3.31B + 0.8370C − 3.22AB + 0.3552AC + 1.40BC(8)
Hausner Ratio = +1.33 + 0.0640A − 0.0613B + 0.0134C − 0.0606AB + 0.0057AC + 0.0223BC(9)

The concentration of the phenolic extract had the most significant positive influence on both the cohesiveness and flowability of the pomegranate powder (Equations (8) and (9)).

Figure 8 (a1,a2,b1,b2) indicates the interactive effects of the phenolic extract concentration, drying temperature, and feed flow rate on the cohesiveness and flowability of the powder. The CI values of the pomegranate powder ranged from 17.8 to 33.4%, and the HR values were between 1.2 and 1.5, demonstrating that the pomegranate powders had fair to very poor flowability and partly low to very high cohesiveness (Table 5).

Based on the obtained results (Table 1 and Figure 8) as well as the flowability and cohesiveness properties (Table 5), it could be concluded that the pomegranate powders produced at higher drying temperatures, lower feed flow rates, and lower phenolic extract concentrations had better flowability and less cohesiveness. Ong et al. [46] declared that mixed mango powder had a higher cohesive ratio (1.35) and flow index (26.1%) than green and ripe mango powders. Tze et al. [74] indicated that pitaya powder produced by spray drying had poor flowability and high cohesive properties, while Saifullah et al. [75] showed that the powders of four types of fruit, including pitaya, pineapple, mango, and guava, had passable flowability.

#### 3.2.9. Brightness

Color, especially brightness, is an important property for quality evaluation, and the drying conditions and the additional materials are the most important parameters affecting food color changes [76].

The ANOVA results for L* are summarized in Table 2 and indicate that for the L* values of the powder, the phenolic extract concentration had a highly significant effect (*p* < 0.01), and the effect of the interaction between the phenolic extract concentration and feed flow rate was significant (*p* < 0.05). Additionally, the quadratic terms of the three variables were not significant (*p* > 0.05).

This result is in disagreement with those announced by Naji-Tabasi et al. [70] and Rashidi et al. [77], who mentioned that the drying temperature had a significant effect on the color of barberry fruit pulp and tomato powder.

By introducing the linear model into the backward elimination algorithm, the following equation was obtained, composed of the coded factors that had significant effects on the L* of the powder:L* = +88.68 − 1.31A + 0.41B + 0.27C − 0.1125AB + 1.06AC − 0.0625BC + 1.36A² + 1.56B² − 1.54C(10)

Equation (10) describes that the concentration of the pomegranate peel phenolic extract had the most reducing influence on the L* of the powder.

The interactive effect of the phenolic extract concentration and feed flow rate on L* is depicted in Figure 9. L* decreased by increasing the phenolic extract concentration but rose with a rise in the feed flow rate. The values of the L* were between 86.1 and 92.6, showing that the powder brightness was close to white. From this result, it could be concluded that the powder color was affected by the spray drying process, leading to a color that was different from the raw material as a result of enzymatic or non-enzymatic browning [59]. On the other hand, the effects of the drying temperature and feed flow rate were not significant, which could be explained by the increase in the drying rate in the spray drying process and thus the decrease in the drying time for the oxidation of the color pigments including phenolic compounds. This result is in agreement with those of the TPC test. Nur Hanani et al. [15] and Rubilar et al. [78] came to the conclusion that increasing the concentrations of pomegranate peel and grapeseed extracts in edible film led to a decline in L* due to the presence of polyphenols and anthocyanins.

### 3.3. Optimization

The optimal conditions of the production of the pomegranate juice powder enriched with pomegranate peel phenolic compounds with the minimum values of the moisture content, a_w_, hygroscopicity, dissolution time, CI, HI, and L*, in addition to the maximum values of solubility, WAC, and TPC, were found to be the phenolic extract concentration of 10%, the drying temperature of 189.9 °C, and the feed flow rate of 0.63 mL/s. Under these conditions, the values of moisture content, a_w_, solubility, WAC, hygroscopicity, dissolution time, TPC, CI, HR, and L* were calculated as 9.803%, 0.184, 91.025%, 1.745 mL, 27.278%, 20.079 s, 4726.264 mg/100 g, 19.252%, 1.236, and 88.45, respectively.

### 3.4. Antioxidant Activity of the Optimal Powder

The nature and concentration of the polyphenolic compounds are generally responsible for the antioxidant activity of plant extracts [79]. The main technique for inhibiting the oxidative process by antioxidants is free-radical scavenging [80]. Kharchoufi et al. [14] investigated the antioxidant activity on the scavenging radicals of pomegranate peel extract by DPPH and resulted that the water pomegranate peel extract (at 55 °C) showed acceptable antioxidant yield (3497.02 mmol Trolox/g).

In this study, the reducing free radical-scavenging activity of the optimal pomegranate powder enriched with phenolic pomegranate peel extract was investigated by the DPPH assay (Table 6). As mentioned earlier in the Section 3.2.7 increasing the concentration of phenolic pomegranate extract in the obtained powder led to an increase in the total phenolic content, and as a result, 10% extract had the highest phenolic extract content, resequencing the most free-radical scavenging and antioxidant activity, whose amount, as shown in Table 6, reached up to 89.21. This level is considered to be close to the one that belongs to each of the extracts and concentrates separately, whose values were 98.81 and 95.31%, respectively. Kaderides et al. [6] reported that microwave-assisted phenolics pomegranate peel extract showed up to 94% antioxidant activity using the DPPH model system. The ability of antiradical probably returns to the functional group of the extract components including -OH, -COOH, and C=O due to the high phenolic content of the extract, which supplies the ability of free-radical scavenging [81,82]. Similar results were observed by Zhai et al. [81] who demonstrated that polysaccharide pomegranate peel at 0.1–1mg/mL concentrations had free-radical scavenging ability at levels of 0.99 to 93.44%. However, Galaz et al. [54] indicated that the drying process slightly affected the DPPH radical scavenging in pomegranate peels, which is consistent with the slight decrease in the total phenolic content. Kennas et al. [82] concluded that the fortification of yogurt powder with pomegranate peels improved the antiradical capacity of the obtained powder. Tontul and Topuz [83] mentioned that the presence of phenolic components, ascorbic acid, and anthocyanins was a reason for the antioxidant potential of pomegranate leather.

Considering that pomegranate juice powder fortified with phenolic pomegranate peel extract has high antioxidant activity, making it a valuable source of natural antioxidants, this ingredient could be utilized in the creation of nutraceuticals as a novel solution.

## 4. Conclusions

In this study, spray drying was used to produce pomegranate powder with a high TPC. Then, the optimal conditions were determined using RSM. The results showed that the spray drying had no negative effect on the TPC of the powder, which increased by increasing the phenolic extract concentration. In conclusion, it could be said that this method was suitable to produce phenol-rich fruit powder whose antioxidant and antimicrobial properties are useful in medical sciences.

## Figures and Tables

**Figure 1 foods-12-02066-f001:**
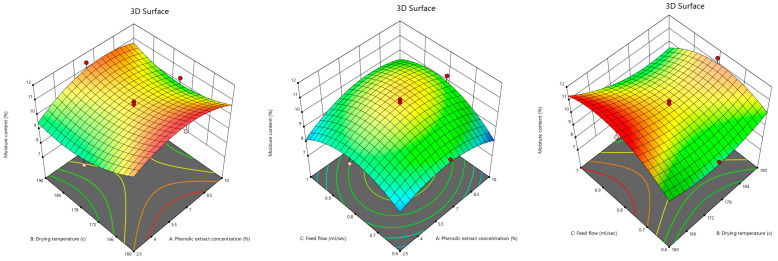
The 3D contour plots illustrating the interactive effects of phenolic extract concentration, drying temperature, and feed flow rate on moisture content of pomegranate powder.

**Figure 2 foods-12-02066-f002:**
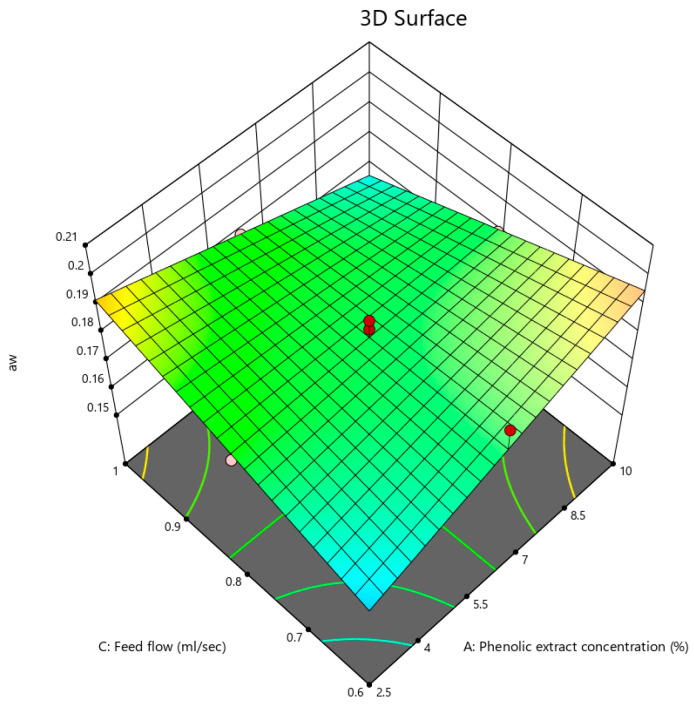
3D contour plots illustrating the interactive effect of phenolic extract concentration and feed flow on a_w_ of pomegranate powder.

**Figure 3 foods-12-02066-f003:**
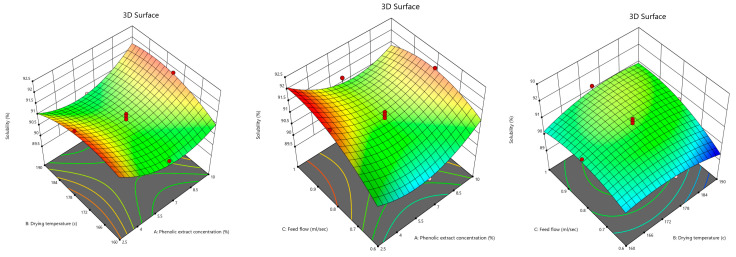
3D contour plots illustrating the interactive effects of phenolic extract concentration, drying temperature, and feed flow rate on solubility of pomegranate powder.

**Figure 4 foods-12-02066-f004:**
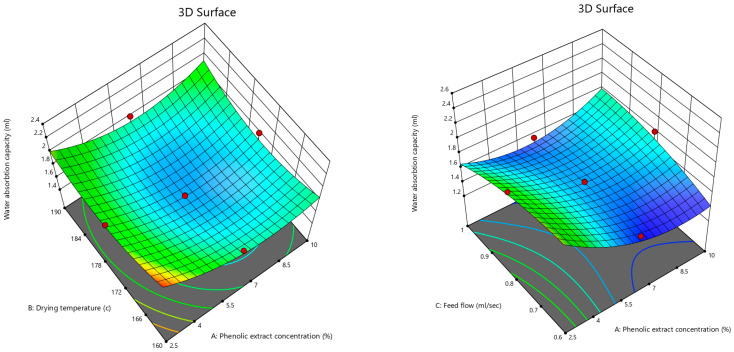
3D contour plots illustrating the interactive effects of phenolic extract concentration and both of the inlet air temperature and feed flow rate on water absorption capacity.

**Figure 5 foods-12-02066-f005:**
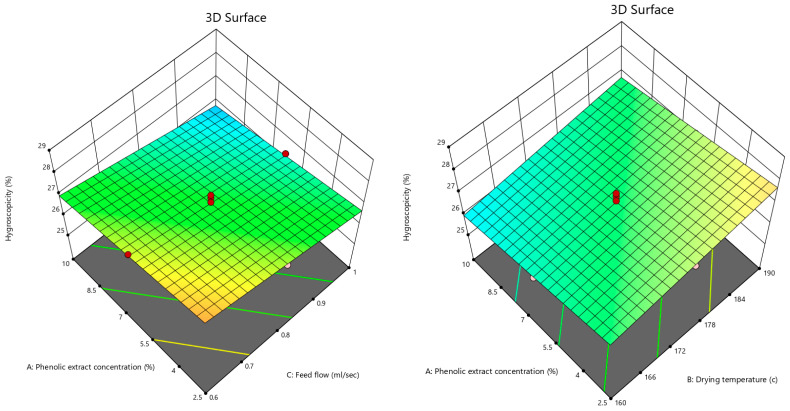
3D contour plots illustrating the interactive effects of phenolic extract concentration and both the inlet air temperature and feed flow rate on hygroscopicity of pomegranate powder.

**Figure 6 foods-12-02066-f006:**
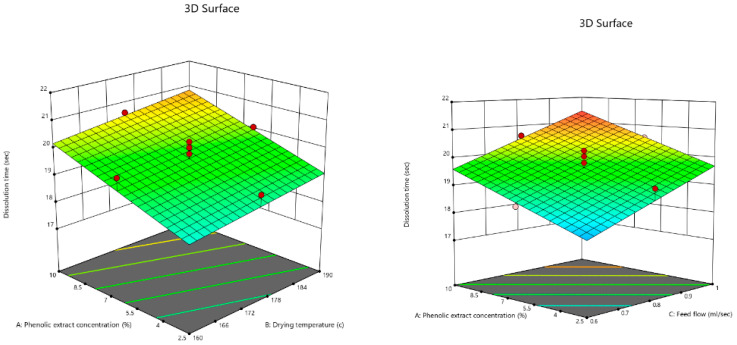
3D contour plots illustrating the interactive effects of phenolic extract concentration and both the inlet air temperature and feed flow rate on the dissolution time of pomegranate powder.

**Figure 7 foods-12-02066-f007:**
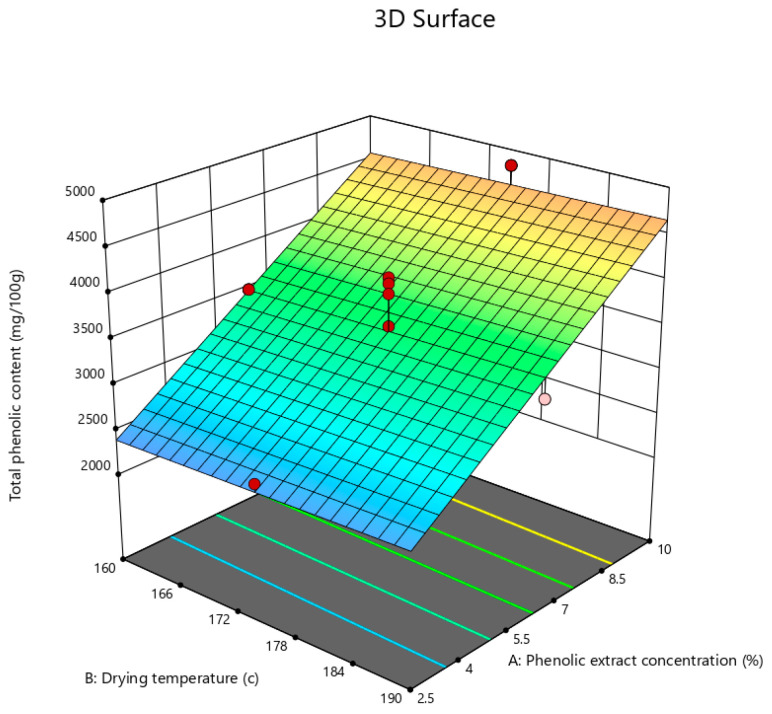
3D contour plot illustrating the effect of phenolic extract concentration on total phenolic content of pomegranate powder.

**Figure 8 foods-12-02066-f008:**
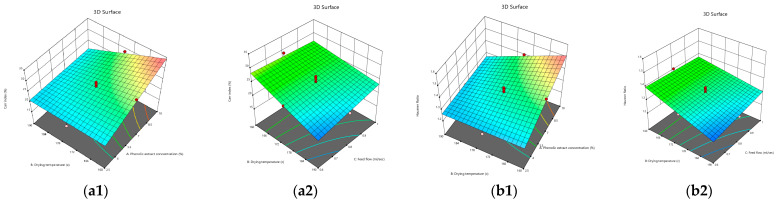
3D contour plots illustrating the interactive effects of phenolic extract concentration and inlet air temperature; as well as inlet air temperature and feed flow rate on flowability (**a1**,**a2**) and cohesiveness (**b1**,**b2**) of pomegranate powder.

**Figure 9 foods-12-02066-f009:**
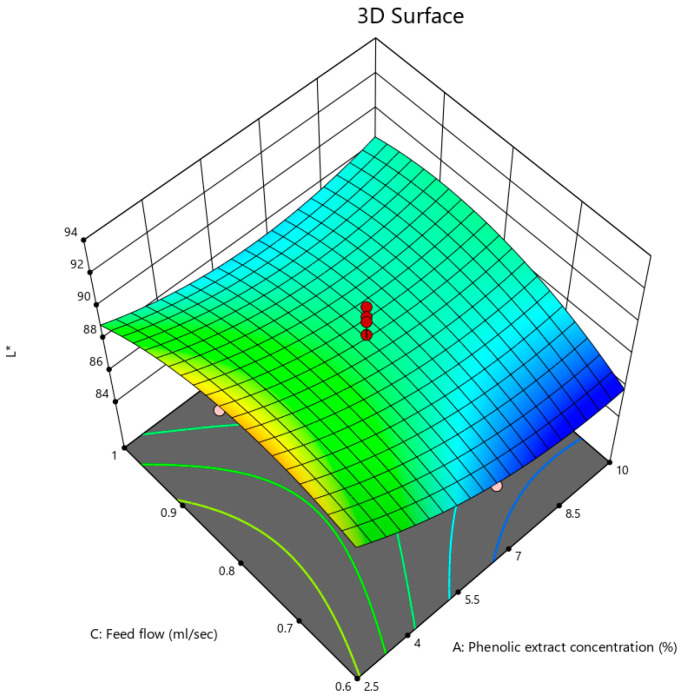
3D contour plot illustrating the interactive effects of phenolic extract concentration and feed flow rate on L* of pomegranate powder.

**Table 1 foods-12-02066-t001:** Central composite face-centered experimental design in coded and actual values and response data.

	Parameters												
Run	A (%)	B (°C)	C (mL/s)	MC (%)	a_w_	SOL (%)	WAC (mL)	HYG (%)	DT (s)	TPC (mg/100 g)	CI (%)	HR	L*
1	10 (+1)	160 (−1)	1 (+1)	10.8317	0.173	90.19	1.97	25.2703	20.81	4365	33.4546	1.5027	90.4
2	6.25 (0)	175 (0)	0.8 (0)	10.3543	0.175	90.799	1.59	26.5616	19.63	4045	26.7269	1.3648	90.6
3	6.25 (0)	175 (0)	0.8 (0)	10.7738	0.176	91.174	1.49	26.8761	19.81	3497	24.9069	1.3317	89.7
4	2.5 (−1)	190 (+1)	1 (+1)	7.839	0.186	91.641	1.67	27.0801	19.95	2170	23.2176	1.3024	91
5	10 (+1)	190 (+1)	0.6 (−1)	9.2337	0.185	90.81	1.66	27.5385	20.23	4976	17.8103	1.2167	87.7
6	10 (+1)	175 (0)	0.8 (0)	10.3585	0.18	91.815	1.8	26.113	20.62	4825	28.7495	1.4035	87.5
7	6.25 (0)	175 (0)	0.8 (0)	10.9457	0.178	91.281	1.51	27.0832	20.25	4111	24.8292	1.3303	87.8
8	2.5 (−1)	175 (0)	0.8 (0)	9.0271	0.175	91.94	1.97	27.2261	19.17	2524	18.9899	1.2344	91.1
9	2.5 (−1)	160 (−1)	0.6 (−1)	9.4829	0.168	90.664	2.25	27.374	17.15	2347	21.3021	1.2707	91.6
10	6.25 (0)	160 (−1)	0.8 (0)	11.3015	0.193	90.8	1.85	26.3846	19.76	3489	28.8955	1.4064	88.4
11	2.5 (−1)	190 (+1)	0.6 (−1)	8.8511	0.154	89.51	2.15	28.2817	17.83	2392	18.9195	1.2333	92.6
12	6.25 (0)	175 (0)	0.8 (0)	10.5839	0.185	90.734	1.58	27.2154	20.04	3932	24.866	1.331	90
13	6.25 (0)	175 (0)	1 (+1)	9.4979	0.179	91.215	1.55	26.3257	20.54	3347	22.6382	1.2926	86.1
14	2.5 (−1)	160 (−1)	1 (+1)	10.3979	0.198	91.59	2.01	26.4672	19.38	2374	20.2664	1.2542	90.6
15	10 (+1)	190 (+1)	1 (+1)	9.5255	0.159	91.71	1.96	26.2961	21.51	4321	23.799	1.3123	90.7
16	6.25 (0)	175 (0)	0.8 (0)	10.1057	0.179	91.032	1.46	26.4221	18.96	3068	25.8327	1.3483	88.1
17	6.25 (0)	190 (+1)	0.8 (0)	11.3852	0.171	90.66	1.8	27.092	20.07	3339	20.4218	1.2566	90.6
18	6.25 (0)	175 (0)	0.8 (0)	10.8742	0.182	90.771	1.52	26.7562	19.37	3578	26.2921	1.3567	88.9
19	10 (+1)	160 (−1)	0.6 (−1)	8.2601	0.201	90.735	1.51	26.6803	19.11	4197	33.339	1.5001	87.5
20	6.25 (0)	175 (0)	0.6 (−1)	9.3279	0.186	90.053	1.55	27.5057	18.57	3551	23.6346	1.3095	86.7

MC: moisture content; SOL: solubility; WAC: water absorption capacity; HYG: hygroscopicity; DT: dissolution time.

**Table 2 foods-12-02066-t002:** Results of analysis of variance (ANOVA) of quadratic model for moisture content, solubility, water absorption capacity, and L* of powder.

		Moisture Content				Solubility				Water Absorption Capacity				L*			
Source	df	Sum of Squares	Mean Square	F Value	*p* Value	Sum of Squares	Mean Square	F Value	*p* Value	Sum of Squares	Mean Square	F Value	*p* Value	Sum of Squares	Mean Square	F Value	*p* Value
Model	9	17.58	1.95	13.59	0.0002	6.80	0.7559	17.29	<0.0001	1.04	0.1158	19.66	<0.0001	47.87	5.32	3.83	0.0240
A-Phenolic extract concentration	1	0.6820	0.6820	4.75	0.0544	0.0007	0.0007	0.0165	0.9003	0.1322	0.1322	22.45	0.0008	17.16	17.16	12.34	0.0056
B-Drying temperature	1	1.18	1.18	8.23	0.0167	0.0124	0.0124	0.2834	0.6061	0.0123	0.0123	2.08	0.1798	1.68	1.68	1.21	0.2973
C-Feed flow	1	0.8622	0.8622	6.00	0.0343	2.09	2.09	47.85	<0.0001	0.0002	0.0002	0.0272	0.8724	0.7290	0.7290	0.5243	0.4856
AB	1	1.02	1.02	7.10	0.0237	0.9099	0.9099	20.81	0.0010	0.0421	0.0421	7.14	0.0234	0.1012	0.1012	0.0728	0.7928
AC	1	1.10	1.10	7.62	0.0201	0.9126	0.9126	20.87	0.0010	0.2738	0.2738	46.48	<0.0001	9.03	9.03	6.50	0.0289
BC	1	2.21	2.21	15.39	0.0029	0.8778	0.8778	20.08	0.0012	0.0200	0.0200	3.40	0.0952	0.0312	0.0312	0.0225	0.8838
A²	1	2.18	2.18	15.14	0.0030	1.53	1.53	35.09	0.0001	0.1669	0.1669	28.34	0.0003	5.11	5.11	3.68	0.0841
B²	1	1.59	1.59	11.09	0.0076	0.4413	0.4413	10.09	0.0099	0.0955	0.0955	16.21	0.0024	6.72	6.72	4.84	0.0525
C²	1	3.76	3.76	26.16	0.0005	0.6782	0.6782	15.51	0.0028	0.0216	0.0216	3.67	0.0845	6.49	6.49	4.67	0.0560
Residual	10	1.44	0.1437			0.4372	0.0437			0.0589	0.0059			13.90	1.39		
Lack of Fit	5	0.9076	0.1815	1.71	0.2844	0.1707	0.0341	0.6402	0.6818	0.0460	0.0092	3.55	0.0954	7.80	1.56	1.28	0.3978
Pure Error	5	0.5296	0.1059			0.2666	0.0533			0.0130	0.0026			6.11	1.22		
Cor Total	19	19.02				7.24				1.10				61.77			

**Table 3 foods-12-02066-t003:** Results of analysis of variance (ANOVA) of 2FI model for water activity, Carr index, and Hausner ratio of powder.

		Water Activity				Carr Index				Hausner Ratio			
Source	df	Sum of Squares	Mean Square	F Value	*p* Value	Sum of Squares	Mean Square	F Value	*p* Value	Sum of Squares	Mean Square	F Value	*p* Value
Model	6	0.0023	0.0004	27.19	<0.0001	334.84	55.81	25.42	<0.0001	0.1139	0.0190	34.78	<0.0001
A-Phenolic extract concentration	1	0.0000	0.0000	2.03	0.1779	118.73	118.73	54.09	<0.0001	0.0410	0.0410	75.07	<0.0001
B-Drying temperature	1	0.0006	0.0006	42.72	<0.0001	109.49	109.49	49.88	<0.0001	0.0376	0.0376	68.76	<0.0001
C-Feed flow	1	1.000 × 10^−7^	1.000 × 10^−7^	0.0070	0.9345	7.01	7.01	3.19	0.0973	0.0018	0.0018	3.28	0.0931
AB	1	2.000 × 10^−6^	2.000 × 10^−6^	0.1404	0.7139	82.90	82.90	37.76	<0.0001	0.0294	0.0294	53.75	<0.0001
AC	1	0.0017	0.0017	118.10	<0.0001	1.01	1.01	0.4599	0.5096	0.0003	0.0003	0.4760	0.5024
BC	1	2.000 × 10^−6^	2.000 × 10^−6^	0.1404	0.7139	15.70	15.70	7.15	0.0191	0.0040	0.0040	7.30	0.0181
Residual	13	0.0002	0.0000			28.54	2.20			0.0071	0.0005		
Lack of Fit	8	0.0001	0.0000	1.01	0.5211	25.12	3.14	4.60	0.0548	0.0060	0.0007	3.34	0.1002
Pure Error	5	0.0001	0.0000			3.41	0.6826			0.0011	0.0002		
Cor Total	19	0.0025				363.37				0.1210			

**Table 4 foods-12-02066-t004:** Results of analysis of variance (ANOVA) of 2FI model for hygroscopicity, dissolution time, and total phenolic content of powder.

		Hygroscopicity				Dissolution Time				Total Phenolic Content			
Source	df	Sum of Squares	Mean Square	F Value	*p* Value	Sum of Squares	Mean Square	F Value	*p* Value	Sum of Squares	Mean Square	F Value	*p* Value
Model	3	7.27	2.42	57.55	<0.0001	17.54	5.85	46.05	<0.0001	1.193 × 10^7^	3.976 × 10^6^	40.97	<0.0001
A-Phenolic extract concentration	1	2.05	2.05	48.74	<0.0001	7.74	7.74	61.00	<0.0001	1.183 × 10^7^	1.183 × 10^7^	121.90	<0.0001
B-Drying temperature	1	1.69	1.69	40.14	<0.0001	1.14	1.14	9.00	0.0085	1.8147 × 10^4^	1.8147 × 10^4^	0.1870	0.6712
C-Feed flow	1	3.53	3.53	83.78	<0.0001	8.65	8.65	68.13	<0.0001	7.8499 × 10^4^	7.8499 × 10^4^	0.8088	0.3818
Residual	16	0.6740	0.0421			2.03	0.1269			1.553 × 10^6^	9.7052 × 10^4^		
Lack of Fit	11	0.2161	0.0196	0.2145	0.9843	0.9427	0.0857	0.3937	0.9082	7.557 × 10^5^	6.8701 × 10^4^	0.4309	0.8864
Pure Error	5	0.4579	0.0916			1.09	0.2177			7.971 × 10^5^	1.594 × 10^5^		
Cor Total	19	7.95				19.57				1.348 × 10^7^			

**Table 5 foods-12-02066-t005:** Classification of flowability and cohesiveness of powder [73].

Carr Index (%)	Flowability	Hausner Ratio
<10	Excellent	1–1.11
11–15	Good	1.12–1.18
16–20	Fair	1.19–1.25
21–25	Passable	1.26–1.34
26–31	Poor	1.35–1.45
32–37	Very poor	1.46–1.59
>38	Very, very poor	>1.6

**Table 6 foods-12-02066-t006:** Antioxidant activity of phenolic pomegranate peel extract, pomegranate concentrate, and optimal pomegranate powder.

Sample	Phenolic Extract Concentration (%)	Inlet Drying Temperature (°C)	Feed Flow Rating (ml/s)	Total Phenolic Content (mg/100 g)	DPPH (%)
Phenolic pomegranate peel extract	―	―	―	3515.5	98.81
Pomegranate concentrate	―	―	―	9230	95.31
Optimal pomegranate powder	10	189.9	0.63	4976	89.21

## Data Availability

The data presented in this study are available on request from the corresponding author.

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
