# Peer review of "Evaluation of Effects of Spray Drying Conditions on Physicochemical Properties of Pomegranate Juice Powder Enriched with Pomegranate Peel Phenolic Compounds: Modeling and Optimization by RSM"

_foods, 2023, doi:10.3390/foods12102066_

Round 1

Reviewer 1 Report

I have reviewed manuscript " Evaluation of effects of spray drying conditions on physico chemical properties of pomegranate juice powder enriched with pomegranate peel phenolic compounds: Modeling and Optimization by RSM ". However, there are some problems in the paper needed to be clarified, which are shown in the following comments:

1.      The conclusions of the study should be Clarified in the abstract.

2.      The novelty of the research should be emphasized in the introduction.

3.      Line 59-64 should be rewritten.

4.      How the concentration of pomegranate peel extract was determined?

5.      How the inlet air temperature (160-190 ° C ) and feed flow rate ( 0.6-91 ml / sec ) were determined in line  91, or add ' on the basis of pre-treatment '.

6.      Please check if the expression of formula correct in the 114,133 and 134, and what is the unit of variable?

7.      The corresponding formula should be added in 2.3.5, 2.3.6 and 2.3.7 sections.

8.      The corresponding references should be added in 2.3.9 chapter.

9.      Results should be rewritten. please follow any other published paper format for result writing.

10.   It is better to write the discussion section separately from the conclusion and expand it with previous research. Future research directions may also be highlighted.

Author Response

Responses to the comments of referees

Detailed reviewer#1 comments

  1. The conclusions of the study should be Clarified in the abstract.

Response:

Thank you very much for your comments to make this manuscript as comprehensive as possible. It has been done, please kindly see lines 26 - 30.

  1. The novelty of the research should be emphasized in the introduction.

Response:

Thanks for your attention. It has been done, please kindly see lines 66 - 76.

  1. Line 59-64 should be rewritten.

Response:

Thanks for your attention. It has been done, please kindly see lines 76 - 81.

  1. How the concentration of pomegranate peel extract was determined?

Response:

Thanks for your attention. The concentration of pomegranate peel extract was determined based on pre-treatment by sensory evaluation of the obtained powder so that the highest possible content of phenolic compounds was added and did not add an astringent taste to the resulting pomegranate powder, in addition to having acceptable flow and adhesion properties for the resulting powder.

  1. How the inlet air temperature (160-190 ° C ) and feed flow rate ( 0.6-91 ml / sec ) were determined in line  91, or add ' on the basis of pre-treatment '.

Response:

Thanks. It has been done, please kindly see line 109.

  1. Please check if the expression of formula correct in the 114,133 and 134, and what is the unit of variable?

Response:

Thank you very much for your interesting suggestion. They have been done, please kindly see sections 2.3.4 and 2.3.8.

  1. The corresponding formula should be added in 2.3.5, 2.3.6 and 2.3.7 sections.

Response:

Thanks for your attention. This has now been modified, please kindly see sections 2.3.5, 2.3.6 and, 2.3.7. 

  1. The corresponding references should be added in 2.3.9 chapter.

Response:

Thanks for your attention. This has now been modified, please kindly see section 2.3.9.

  1. Results should be rewritten. please follow any other published paper format for result writing.

Response:

Thanks for your attention. This has now been modified, please kindly see it.

  1. It is better to write the discussion section separately from the conclusion and expand it with previous research. Future research directions may also be highlighted.

Response:

Highlights has been added. Please kindly see lines 649-657.

Reviewer 2 Report

In this manuscript, the spray drying conditions (pomegranate peel extract concentration, drying temperature and feed flow rate) of pomegranate juice powder were optimized by RSM. 

Comments:

1. Why are these factors selected? Other factors, e.g. brix of pomegranate juice, DE value of maltodextrin, can also affect the physicochemical properties of pomegranate juice powder.

2. The more pomegranate peel extract was added, the high total phenolic content was obtained in the pomegranate juice powder. How to determine the tested pomegranate peel extract concentration at 2.5-10%?

Author Response

Detailed reviewer#2 comments

  1. Why are these factors selected? Other factors, e.g. brix of pomegranate juice, DE value of maltodextrin, can also affect the physicochemical properties of pomegranate juice powder.

Response:

Thank you very much for your comments to make this manuscript as comprehensive as possible. So right, these factors definitely can affect the properties of the powder, but the main aim of this study was enriching the pomegranate juice powder with phenolic compounds and investigation the effect of spray drying conditions on the preservation of these compounds; so other factors was less important for our research in compared with feed flow rate and drying temperature which can destroy or delete their antioxidant and antimicrobial properties.

  1. The more pomegranate peel extract was added, the high total phenolic content was obtained in the pomegranate juice powder. How to determine the tested pomegranate peel extract concentration at 2.5-10%?

Response:

Thanks for your attention. The concentration of pomegranate peel extract was determined based on pre-treatment by sensory evaluation of the obtained powder so that the high possible content of phenolic compounds was added and did not add an astringent taste to the resulting pomegranate powder, in addition to having acceptable flow and adhesion properties for the resulting powder.

Reviewer 3 Report

Although the work under the report is of significance but needs major revision for this manuscript. I have highlighted or cut the lines for raised objections. The major concern is that fitting different models for a group of parameters do not make scientific sense. I do not understand why MD (de16) was used specifically. The authors also do not explain that. In materials, all the chemicals are not listed as per the methodology. Mixing juice concentrate with peel concentrate for health benefits is good but the whole process goes futile if the critical parameters like antioxidant capacity, suppression of unwanted free radical reactions e.g. DNA damage repair, etc through various in vitro assays are not showcased with this work.

It needs language editing.

Author Response

Detailed reviewer#3 comments

  1. Although the work under the report is of significance but needs major revision for this manuscript.

Response:

Thank you very much indeed for your positive feedback and we are grateful for the time and energy you expended on our manuscript. We have considered all your suggestions/comments and modified the manuscript as much as possible.

  1. I have highlighted or cut the lines for raised objections.

Response:

Thank you very much indeed for your meticulous comments.

  • Line 32 has been modified, please kindly see lines 35.
  • The immunity word in line 42 has been modified in the revised manuscript at line 45. According to Karimi et al., (2017) the high amount of bioflavonoids present in pomegranate is considered as a remedy to cure acquired immune deficiency syndrome (AIDS) owing to its free radical scavenging and lipoxygenase inhibitory effects. Health promoting compounds present in pomegranate contribute to its remedial properties like hypolipidemic, antiviral, anti-neoplastic, helminthic, digestive protection, and immunomodulation activities.

Reference:

Karimi, M., Sadeghi, R., & Kokini, J. Pomegranate as a promising opportunity in medicine and nanotechnology. Trends in Food Science & Technology. 2017, 69, 59–73.

  • The sentence in lines 223 and 224 have been rewritten in the revised manuscript at lines 254 and 255. Please kindly see it.
  • The line 229 has been corrected in the revised manuscript at line 260.
  • The sentence in line 245 have been rewritten in the revised manuscript at lines 276 and 277. Please kindly see it.
  1. The major concern is that fitting different models for a group of parameters do not make scientific sense.

Response:

  1. I do not understand why MD (de16) was used specifically. The authors also do not explain that.

Response:

Thanks for your attention. It has been added in lines 61-65. Please kindly see it.

Maltodextrin is starch hydrolysis product with a dextrose equivalent (DE) of less than 20 and high solubility in water. It is commonly used as a bulking agent, stabilizer, and thickener in food and beverage industries. In powder producing, maltodextrin is used as a carrier or filler for dry ingredients to improve flowability and prevent clumping. It also helps to enhance the texture, mouthfeel, and shelf life of powders. Moreover, maltodextrin can be easily spray-dried to produce fine powder particles with good dispersibility and instant solubility. Therefore, it is an ideal choice for the production of various powders, specifically sugar and acid rich fruit juice powders (Vidović et al., 2014).

The DE value of maltodextrin varies with changes in the chain lengths of sugar molecules. The shorter is the average chain length, the higher is the DE value. Moreover, maltodextrins with different DE values have different properties and functions in microcapsules. Matsuura et al., (2015) used maltodextrins with DE values of 2, 10, and 25 as wall materials because of differences in the composition and structure of maltodextrin and in its interactions with emulsifiers. Although maltodextrin with a DE value of 10 had the strongest effects with emulsifiers, the powder formed by spray drying had the lowest stability. However, high DE values can improve the stability of the core materials in the spray drying process because that the matrices are more uniform after drying, which thereby increases the retention rate and produces a powder with low hygroscopicity (Ghani et al., 2017). Further, the solubility of maltodextrin increases with an increase in the DE value, which may accelerate the dissolution of powder microcapsules (Li et al., 2020). Vidović et al., (2014) used the maltodextrin (DE16) as a carrier of health benefit compounds in Satureja montana dry powder extract obtained by spray drying technique. Igual et al., (2014) in their research used maltodextrin (DE16.5-19.5) for produce lulo (Solanum quitoense L.) pulp powder by different spray drying conditions. Cano-Chauca et al., (2005) studied the effect of various carriers on the functional properties of spray-dried mango powder and concluded that powder solubility increases in function of maltodextrin concentration.

So, the purpose of adding maltodextrin (DE16) in our research is to improve the powder properties, flow and adhesive properties e.g., whose DE was available in the native market. References:

Vidović, S.S., Vladić, J.Z., Vaštag, Ž. G., Zeković, Z.P., & Popović, L.M. 2014. Maltodextrin as a carrier of health benefit compounds in Satureja montana dry powder extract obtained by spray drying technique. Powder Technology, 258, 209–215.

Cano-Chauca, M.; Stringheta, P.C.; Ramos, A.M.; Cal-Vidal, J. 2005. Effect of the carriers on the microstructure of mango powder obtained by spray drying and its functional characterization. Innovative Food Science and Emerging Technologies, 6, 420–428.

Matsuura, T.; Ogawa, A.; Tomabechi, M.; Matsushita, R.; Gohtani, S.; Neoh, T.L.; Yoshii, H. 2015. Effect of dextrose equivalent of maltodextrin on the stability of emulsified coconut-oil in spray-dried powder. Food Engineering, 163, 54–59.

Ghani, A.A.; Adachi, S.; Shiga, H.; Neoh, T.L.; Adachi, S.; Yoshii, H. 2017. Effect of different dextrose equivalents of maltodextrin on oxidation stability in encapsulated fish oil by spray drying. Bioscience Biotechnology Biochemistry, 81, 705–711.

Li, K., Pan, B., Ma, L., Miao, S., and Ji, J. 2020. Effect of Dextrose Equivalent on Maltodextrin/Whey Protein Spray-Dried Powder Microcapsules and Dynamic Release of Loaded Flavor during Storage and Powder Rehydration. Foods journal, 9(1878), 1-18.

Igual, M., Ramires, S., Mosquera, L.H., Martínez-Navarrete, N. 2014. Optimization of spray drying conditions for lulo (Solanum quitoense L.) pulp. Powder technology, 256, 233-238.

  1. In materials, all the chemicals are not listed as per the methodology.

Response:

Thanks very much for your notice. The missing materials have been added, please kindly see the section 2.1.

  1. Mixing juice concentrate with peel concentrate for health benefits is good but the whole process goes futile if the critical parameters like antioxidant capacity, suppression of unwanted free radical reactions e.g. DNA damage repair, etc through various in vitro assays are not showcased with this work.

Response:

Thank you very much for your suggestion. So right, the DPPH test of the optimal pomegranate powder enriched with phenolic pomegranate peel extract has been added to investigation the antioxidant activity of the powder, please kindly see it in section 2.3.10 and 3.4.

  1. It needs language editing.

Response:

Thank you very much for your suggestion. The language of the article was generally improved and some sentences were rewritten. Thanks for the careful comment.

Round 2

Reviewer 1 Report

I recommend accepting in the present form.

Reviewer 2 Report

As the authors have revised the manuscript as suggested, it can be accepted for publication.

Reviewer 3 Report

Congratulations!